# A 3D-Printed Bi-Material Bragg-Based Reflectarray Antenna

**DOI:** 10.3390/s24206512

**Published:** 2024-10-10

**Authors:** Walid Chekkar, Jerome Lanteri, Tom Malvaux, Julien Sourice, Leonardo Lizzi, Claire Migliaccio, Fabien Ferrero

**Affiliations:** 1Laboratory of Electronics, Antennas and Telecommunications (LEAT), CNRS, Université Côte d’Azur, Sophia Antipolis, 06903 Valbonne, France; walid.chekkar@univ-cotedazur.fr (W.C.); jerome.lanteri@univ-cotedazur.fr (J.L.); claire.migliaccio@univ-cotedazur.fr (C.M.); 2Nanoe, 91160 Ballainvilliers, France; t.malvaux@nanoe.com (T.M.); j.sourice@nanoe.com (J.S.); 3Department of Information Engineering and Computer Science, University of Trento, 38123 Trento, Italy; leonardo.lizzi@unitn.it

**Keywords:** 3D-printing, reflectarray, millimeter wave, dual-band, photonic-bandgap, Bragg mirror

## Abstract

This paper presents a 3D-printed fully dielectric bi-material reflectarray with bandgap characteristics for multi-band applications. To achieve bandgap characteristics, a “1D Bragg reflector” unit cell is used. The latter is a layered structure characterized by a spatial distribution of refractive index that varies periodically along one dimension. By appropriately selecting the dimensions, the bandgap can be shifted to cover the desired frequency bands. To validate this bandgap characteristic, a (121.5 mm × 121.5 mm) with an f/D ratio of 0.5 reflectarray was fabricated. The measured gain at 27 GHz is 27.22 dBi, equivalent to an aperture efficiency of 35.05%, demonstrating good agreement between simulated and measured performances within the frequency range of 26–30 GHz. Additionally, the transparency of the reflectarray was verified by measuring the transmission coefficient, which exhibited a high level of transparency of 0.32 dB at 39 GHz. These features make the proposed reflectarray a good candidate for multi-band frequency applications.

## 1. Introduction

Microstrip reflectarrays offer several advantages in various applications due to their unique design and properties, such as high gain, low profile, good efficiency, and beam scanning capability [1]. In contrast to naturally broadband parabolic reflectors, reflectarrays exhibit a limited bandwidth, typically less than ten percent [1]. Despite this limitation, reflectarrays are widely used for multi-band applications [2]. Several approaches have been proposed in the literature to realize dual-band/multi-band reflectarrays.

For relatively close frequencies, the use of broadband [3] or multi-layer elements increases the bandwidth sufficiently to cover both bands. In [4], a three-layer element of varying-sized patch elements was proposed to cover both Tx (11.7–12.2 GHz) and Rx (13.75–14.25 GHz) bands of the Ku-band. However, this solution is unsuitable for widely separated bands. Another approach consists of combining two phasing elements at different bands arranged on a single aperture [5,6,7]. It should be noted that the elements must be spaced enough to reduce the mutual coupling, which affects the efficiency and bandwidth performances of the antenna.

In the case of widely separated frequencies, the low-frequency elements are larger than the high-frequency elements, which makes the physical separation between the bands challenging. To overcome this limitation, FSS-backed multilayer reflectarray structures have been proposed in [2,8,9]. A frequency-selective surface (FSS), also referred to as a spatial filter, is a periodic structure with bandgap characteristics [10]. The FSS is designed to have maximum reflection at the center frequency of the first band and maximum transmission at the second band. This technique provides good isolation between the reflectarrays of different bands but the profile and dielectric loss of the reflectarray increases as well as the fabrication complexity and cost.

Analogous to frequency-selective surfaces (FSSs), photonic bandgap materials are periodic arrangements of dielectrics with different refractive indexes in a 1D, 2D, or 3D manner, that lead to the appearance of photonic bandgaps, preventing electromagnetic waves at specific wavelengths from propagating while out of the gap the structure is transparent [11]. This bandgap characteristic enables the realization of a fully dielectric dual-band reflectarray and makes its fabrication through 3D-printing possible. To the best of our knowledge, only a few works based on 3D-printing fabrication have been reported at millimeter waves. In [12], the authors applied this principle to design a parabolic-shaped reflectarray antenna in the k and v bands.

The use of 3D-printing, also known as additive manufacturing, has emerged as a promising technique for RF system design. This method offers several advantages over traditional manufacturing methods, including a smaller form factor, lighter weight, lower cost, and eco-friendliness [13]. Furthermore, this technology enables the creation of complicated structures with high precision and increased design flexibility [13]. High -performance systems have been produced using 3D-printing, such as transmitarrays [14,15,16], lenses [17,18], and reflectarrays [12,19,20].

In this paper, a 3D-printed bi-material reflectarray antenna is presented. The antenna is built by periodically stacking dielectric layers with different permittivity using two different materials. It is noteworthy that 3D-printing allows the use of bi-material technology through the dual extruder configuration of certain 3D-printers. This configuration enables the printing of the entire structure in a single print without the need for supports to fix the different layers, resulting in a much more rigid structure.

Compared to [12], the innovation in this work lies in the use of the bi-material configuration, which enables great versatility in permittivity by modifying the porosity through 3D-printing. This allows for a shorter focal length and, consequently, results in a much more compact system, which is crucial for 5G base stations due to the need for space efficiency, ease of installation, and performance optimization. To simplify the reflectarray antenna design, four unit cells were optimized for 2-bit phase quantization. Taking advantage of its stepped configuration, the reflectarray exhibits bandgap characteristics, which make dual-band properties possible. Based on this concept, this paper aims to provide practical evidence of these bandgap characteristics by measuring both the reflection and transmission of a bi-material reflectarray.

This paper is organized as follows: the bandgap characteristics of the dielectric reflectarray element and its phase response are theoretically investigated in Section 2. Section 3 describes the measurement procedure and discusses the reflectarray performances. Lastly, conclusions and perspectives are discussed in Section 4.

## 2. Fully Dielectric Reflectarray Antenna Design

### 2.1. Bragg Reflector-Based Unit Cell

The unit cell is a “1D Bragg reflector”, which is a layered structure characterized by a spatial distribution of refractive index that varies periodically along one dimension [21]. This can be achieved either by using two different materials or by changing the porosity of a single material.

The principle of operation of a Bragg mirror relies on the fundamental principle of the Bragg law to achieve constructive interference in reflection and destructive interference in transmission. The thickness of each layer equals a quarter guided wavelength ($\lambda_{g}$) for normal incidence. At each interface within the mirror, a Fresnel reflection occurs as a result of the difference in refractive index between the layers. For the design wavelength, the incident beam undergoes a phase shift of −π/2 at each passage through a dielectric layer. Additionally, the wave propagating from a low-index to a high-index medium undergoes a −π phase shift (while a 0 phase shift is obtained for the opposite case). The reflected waves emerge in phase, leading to constructive interference. A simple equation can be used to calculate the reflectivity at normal incidence:(1)$$R_{max}={\left(\frac{1-{\left(\frac{n_{1}}{n_{2}}\right)}^{2N}}{1+{\left(\frac{n_{1}}{n_{2}}\right)}^{2N}}\right)}^{2}$$
where *N* is the number of the quarter wave pairs, and $n_{1}$ and $n_{2}$ are the refractive indexes of the two layers. The mirror’s reflectivity depends on both the refractive index contrast between the materials and the number of alternating layer pairs. Ideally, the number of layers is infinite but sufficient reflection can be achieved using multiple periodicities with a large index difference $\Delta_{n}$, as shown in Figure 1.

The stop-band, which refers to the range of reflected frequencies, can be calculated by [22]:(2)$$\Delta \lambda_{max}=\frac{4\lambda_{0}}{\pi }sin^{-1}\left(\frac{\Delta_{n}}{n_{1}+n_{2}}\right)$$
where $\lambda_{0}$ is the free space wavelength, $\Delta \lambda_{max}$ is the reflected wavelength range, and $\Delta_{n}$ is the index contrast. Equation (Equation 2) demonstrates that a larger stop-band can be achieved by utilizing a high refractive index contrast.

In this work, the Bragg mirror’s geometry involves six layers of dielectric material, each 1.5mm thick, with a relative permittivity of 7.5 and a loss tangent of 0.001. A dielectric layer with a dielectric constant of 1.6 is placed in between. The latter is constituted by 60% infill density of another dielectric material with a permittivity of 2.2, a thickness of 1.58 mm, and a loss tangent of 0.001. These materials, part of the Zetamix Epsilon series developed by Nanoe [23], are high-purity ceramics developed for industrial applications characterized at 9.4 GHz. Thanks to their polyolefin composition, these materials can be used at higher frequencies compared to alternative materials, as evidenced by a notable 110 °C heat deflection temperature for the Zetamix Epsilon series versus a maximum of 70–80 °C for filled ABS. The material permittivity as a function of the porosity level can be calculated using the Maxwell–Garnett equation [24]:(3)$$\epsilon =\epsilon_{2}\frac{\epsilon_{1}+2\epsilon_{2}+2f\left(\epsilon_{1}-\epsilon_{2}\right)}{\epsilon_{1}+2\epsilon_{2}-f\left(\epsilon_{1}-\epsilon_{2}\right)}.$$
Here, ϵ denotes the new permittivity, $\epsilon_{1}$ represents the intrinsic permittivity of the material, and $\epsilon_{2}$ stands for the permittivity of air. *f* represents the volume fraction of the scatterer phase in the mixture.

The thickness of the layers was carefully chosen to maximize reflection at 27 GHz and optimize transmission at 39 GHz, thus enabling coverage of both 5G bands by stacking two reflectarrays at these frequencies. This design not only improves performance in these specific frequency bands, but also facilitates the process of 3D-printing of the reflectarray, ensuring ease of manufacture and functionality in the intended frequency ranges.

A parametric study showed that the frequency band in which the unit cell remains transparent widens as more layers are applied. However, the thickness of the reflectarray also increases. A choice of 6 layers was made, representing a good trade-off between transparency and reflectarray thickness.

The unit cell frequency response was simulated using the commercial full-wave solver CST Microwave Studio with unit cell boundary conditions and Floquet port excitation and normal incidence. The simulation model and its reflection and transmission magnitude are represented in Figure 2a. As can be seen from Figure 2b, a bandgap extending from 18 GHz to 31 GHz has formed. The reflection coefficient is close to 0 dB within this band, while it is less than −10 dB from 32.3 GHz to 44.2 GHz, indicating the reflection and transmission bands of the unit cell. It is important to note that several bandgaps have appeared, but only the one that covers the selected frequency band is considered.

It is beneficial at this point to check how the proposed unit cell responds to changes in both the angle of incidence and polarization state. We investigated the behavior of the unit cell under different polarization states as the angle of incidence varied. Specifically, three cases were examined: TE polarization, TM polarization, and linear polarization at 45°, which represents a linear combination of TE and TM modes. We consider a system composed of alternating dielectric layers with wave propagation along the *z*-axis and the incident plane in the *x-y* plane (Figure 2a). This configuration defines a symmetry plane, allowing us to distinguish between two independent electromagnetic modes: transverse electric (TE) and transverse magnetic (TM) modes. For the TE mode, the electric field E is perpendicular to the *x-y* plane, meaning it is aligned along the *z*-axis, and the magnetic field H lies in the *x-y* plane. Conversely, for the TM mode, the magnetic field H is perpendicular to the *x-y* plane (along the *z*-axis), while the electric field E lies in the *x-y* plane.

First, we fixed the polarization and varied the angle of incidence. As can be seen from Figure 3b, the location of the bandgap shifts slightly towards higher frequencies as the angle of incidence increases for the TE mode, whereas the bandgap for the TM mode decreases. The unit cell maintains a reflection coefficient close to 0 dB at 27 GHz and a transmission coefficient (Figure 3b) close to 0 dB at 39 GHz, indicating that the unit cell is transparent.

As the angle of incidence approaches the Brewster angle, which in this case is 70°, calculated by:$$\theta_{B}=\arctan\left(\frac{n_{2}}{n_{1}}\right)$$
where $n_{1}$ and $n_{2}$ are the refractive indices of air and the first layer of the unit cell, the TM mode is efficiently transmitted through the structure as expected, while the TE mode still reflects significantly in the frequency range from 20 to 37 GHz, as can be seen in Figure 4a.

Finally, we analyzed the behavior of the unit cell under 45° linear polarization, which combines both TE and TM modes. The amplitude results show that for angles of incidence between 0° and 40°, both TE and TM modes are superposed, exhibiting nearly identical amplitude and phase responses, as shown in Figure 5 and Figure 6. This is due to the fact that at 45° linear polarization, the TE and TM modes are equally excited, and the symmetry of the unit cell ensures that at lower angles of incidence (between 0° and 40°), the interaction between the incident wave and the structure affects both modes in the same way, resulting in almost identical amplitude and phase responses. However, as the angle of incidence increases beyond 40°, the TM mode is increasingly transmitted, particularly in the higher frequency range (30–45 GHz), while the TE mode continues to reflect, as observed in Figure 4b. Notably, good transmission is maintained at 39 GHz. Ensuring that both the electric (E) and magnetic (H) fields are transmitted with equal phase and amplitude is critical in a dual-band reflectarray based on a Bragg mirror. This balance is essential to achieve efficient operation in both frequency bands, thereby optimizing the reflectarray’s performance for multi-band applications.

### 2.2. Reflectarray Design

As the elements should satisfy the desired phase distribution to produce a collimated beam in a certain direction, a reference plane is placed above the Bragg-mirror-type unit cell, as shown in Figure 2a. The reflection phase is evaluated by modifying the position of the element along the z-axis while keeping the same reference plane. A full phase-cycle (2π) is achieved by changing the element position from −1 to 5 mm, as shown in Figure 7a. It is important to note that the phase curve shown here is for a normal incidence. The reflectarray is synthesized using two unit cells (2-bit phase quantization). Thus, the positions of each unit cell are −0.63 mm, 0.7 mm, 2 mm, and 3.35 mm. The required phase shift (represented in Figure 7b) $\Phi (x_{i},y_{i})$ at each element $\left(x_{i},y_{i}\right)$ can be calculated by:(4)$$\Phi \left(x_{i},y_{i}\right)=k_{0}\left(d_{i}-\left(x_{i}\cos\phi_{b}+y_{i}\sin\phi_{b}\right)\sin\theta_{b}\right)$$
where $k_{0}$ is the propagation constant in a vacuum, and $d_{i}$ is the distance between the feed phase center and the *i*-th element. A 27 × 27 reflectarray is designed and simulated. The aperture size is 121.5 mm × 121.5 mm (≈11.34λ). A linearly polarized WR28 waveguide feed is placed on the focal point F = 60.75 mm with an f/D ratio of 0.5.

## 3. Fabrication and Measurement

### 3.1. 3D-Printing with Dual Material

The dielectric Bragg-based reflectarray prototype was printed using a dual extruder 3D-printer Raise 3D Pro 2 from Raise3D Technologies as shown in Figure 8. As the reflectarray consists of two blocks of different permittivities, each sub-block was assigned to an extruder loaded with Zetamix Epsilon 7.5 and 2.2 filaments, respectively. The respective infill densities obtained for both sub-blocks correspond to that given by (Equation 4). To ensure a flat base that facilitates printing, an additional dielectric layer was added to the bottom of the structure, as shown in Figure 8.

The build plate is heated to 110 °C, and both filaments are printed at 270 °C. It is essential to switch off the cooling fans completely to avoid warping and maintain optimum layer adhesion. As the printing begins, the printer alternates between the two extruders, depositing the corresponding filament according to the predefined design. This dual-extrusion capability allows for the simultaneous deposition of two filaments during the printing of a single object, facilitating the creation of multi-material structures.

### 3.2. Radiation Measurement

The radiation performance is measured using the Rohde&Schwarz R&S®ATS800B CATR benchtop antenna test system [25], which provides a highly compact environment for characterizing antennas and 5G devices in the 20 GHz to 50 GHz frequency range. The gain of the open waveguide is 6.5 dBi at 27 GHz. To mitigate the blocking effect, the PLA-based measurement support is 3D-printed with a 10% infill density, corresponding to $\epsilon_{r}$ = 1.09, as determined by (Equation 4).

The measured and simulated radiation patterns at 27 GHz are presented in Figure 9. The results reveal a good agreement between measurement and simulation, indicating a peak measured gain of 27.22 dBi. Furthermore, the measured cross-polarization component is below −35.54 dB, and the measured side lobe level (SLL) is less than −17.27 dB. However, subtle fluctuations are observed. They can be ascribed to the fact that the amounts of undesirable edge diffraction, specular reflection, and feed blockage increase relative to the desirable radiation from all elements as the aperture size decreases [1].

The measured and simulated gain over the frequency band of the proposed reflectarray antenna is presented in Figure 10. The discrepancy between the measurement and the simulation may be attributed to alignment errors between the reflectarray and the waveguide, as well as fabrication tolerances and measurement accuracy, which is +/−1 dB according to the antenna test system specifications [25]. Additionally, the bandwidth of the fabricated reflectarray, defined at −1 dB, is approximately 3.33%, centered at 27.05 GHz. Furthermore, it is important to acknowledge the impact of material properties on reflectarray performance. In fact, retrospective simulations have shown that a slight reduction in the Zetamix Epsilon 7.5 value can improve gain values. This could potentially explain some of the differences observed between measurements and simulations.

The proposed reflectarray exhibits an aperture efficiency of 35.05%. The aperture efficiency is generally determined as the product of various factors, including:Spillover efficiency $\eta_{s}$.Taper efficiency $\eta_{t}$.Phase efficiency $\eta_{p}$.Polarization efficiency $\eta_{x}$.Blockage efficiency $\eta_{b}$.Random error efficiency $\eta_{r}$ over the reflectarray surface.Material loss $\eta_{m}$.

Generally, antennas are designed to have good matching, symmetrical radiation patterns, aligned phase centers, and minimal cross-polarization. The remaining losses are associated with spillover efficiency, taper efficiency, feed blockage, and dielectric losses.

Table 1 presents the loss budget for the proposed reflectarray at 27 GHz. The taper efficiency, which characterizes the uniformity of the amplitude distribution of the feed pattern across the surface of the reflectarray [26], was estimated at 0.1 dB. On the other hand, the spillover efficiency, representing the fraction of the feed pattern that extends beyond the planar surface of the reflectarray, was estimated at 2.7 dB, playing a significant role in the overall losses. In addition, the dielectric loss was calculated as 0.1 dB, which is relatively low, thanks to the high quality of the used materials. Considering the center-fed symmetric configuration of the proposed reflectarray antenna, the feed and its measurement support may block some radiation of the surface and hence degrade the performance by reducing the gain and also increasing cross-polarization. The blockage efficiency was estimated by calculating the ratio of the blocking surface near the focal point to the main reflectarray surface, and its value is 0.56 dB. The feed can be tilted from the center of the reflectarray to avoid the blockage effect.

To enhance the aperture efficiency, optimizing the feed antenna becomes crucial. Specifically, the WR28 waveguide exhibits an asymmetrical radiation pattern in the E and H planes, resulting in non-uniform illumination of the reflectarray on both planes.

This work aims also to demonstrate the bandgap properties of a Bragg reflectarray. The transparency of this reflectarray was verified through a transmission measurement using the CATR R&S® system. A horn antenna is used to evaluate the transparency of the reflectarray. The procedure involved conducting a reference transmission coefficient measurement between the horn antenna and the R&S® ATS800B feed antenna without the reflectarray. Subsequently, the reflectarray was positioned in front of the horn, and a transmission coefficient measurement between the two antennas was performed once again. The measured S12 parameter is illustrated in Figure 11. The measured transmission coefficient is slightly affected, exhibiting a marginal difference of −0.32 dB compared to the reference, confirming the transparency of the proposed reflectarray at 39 GHz.

### 3.3. Comparison with State of the Art

A comparison between the proposed reflectarray and some other reflectarrays is shown in Table 2. As observed, conventional dual-band reflectarrays rely on grounded resonators printed on one or multiple layers. However, these approaches have several shortcomings, notably, intense mutual coupling. Efforts have been made to mitigate mutual coupling by using frequency-selective surfaces (FSSs) or designing sophisticated resonators [3,9]. In [3], a dual-band reflectarray antenna composed of notched rectangular dielectric resonators of variable height was proposed. Although the reflectarray exhibits low mutual coupling, the system is bulky and its manufacturing process remains complex. In [12], a strong reflection is achieved by superposing layers with different refractive indexes, eliminating the need for a full metallic ground and making the design fully dielectric. Although the reflectarray exhibits good bandwidth performance due to its parabolic shape, facilitating its fabrication required adopting a significant f/D ratio to sufficiently flatten the parabolic profile to maintain unit cells between them, thus making the system voluminous. Additionally, to keep the different layers together, the system requires the use of supports, which may compromise its robustness. In contrast, for the proposed reflectarray, the use of bi-material technology through a dual extruder 3D-printer allowed the printing of the antenna in a single print, eliminating the need for supports to fix the different layers that compose it. Moreover, the versatility of permittivity achieved through bi-material 3D-printing allows the reduction in the size of the reflectarray by a factor of 3.88 compared to [12], which is crucial for 5G base station application due to the need for space efficiency, ease of installation, and performance optimization. This approach makes the reflectarray more compact and robust compared to other designs in the literature. Capitalizing on its bandgap characteristics, the reflectarray demonstrates a high level of transparency of 0.32 dB at 39 GHz, ensuring weak mutual coupling and an acceptable aperture efficiency.

## 4. Conclusions

To achieve bandgap characteristics, a Bragg-reflector-inspired reflectarray was studied and analyzed in this paper. The design leverages a 1D dual-material Bragg reflector unit cell, allowing 3D-printing for fabrication. The fabricated antenna was measured in the frequency range of 25–30 GHz, and the measured gain at 27 GHz is 27.22 dBi, equivalent to an aperture efficiency of 35.05%. In addition, the transparency of the reflectarray was verified by measuring the transmission coefficient. The measurement results show that the reflectarray exhibits a high level of transparency of 0.32 dB at 39 GHz. All these features make the proposed reflectarray a good candidate for multi-band applications.

With regard to future perspectives to enhance the overall performance of the reflectarray, several avenues are considered. Firstly, an illuminating feed with a symmetric radiation pattern to optimize aperture efficiency will be considered. This design will be executed in an offset configuration to address blockage issues contributing to the estimated losses outlined in this article. Furthermore, diffraction effects will be accounted for in the design to minimize their impact on the overall performance of the reflectarray. By combining these approaches, it is envisioned to further optimize the reflectarray’s performance, thereby providing a more efficient and versatile solution for wireless communication systems and radar technologies.

## Figures and Tables

**Figure 1 sensors-24-06512-f001:**
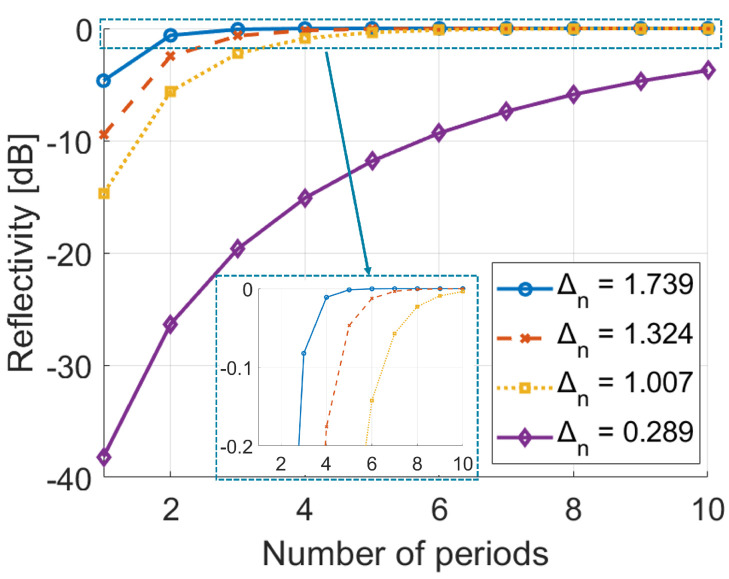
Influence of number of layers and index difference.

**Figure 2 sensors-24-06512-f002:**
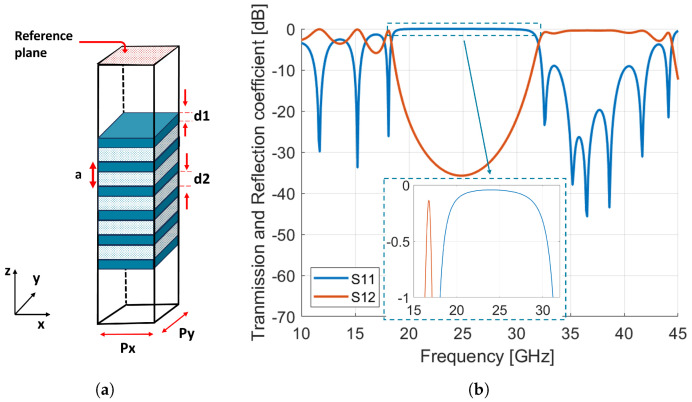
(**a**) Simulation model of dielectric mirror structure in CST microwave studio for a Ka-band reflectarray. (**b**) Transmission and reflection coefficients of the structure. (d1 = 1.5 mm, d2 = 1.58 mm, Px = Py = 4.5 mm.

**Figure 3 sensors-24-06512-f003:**
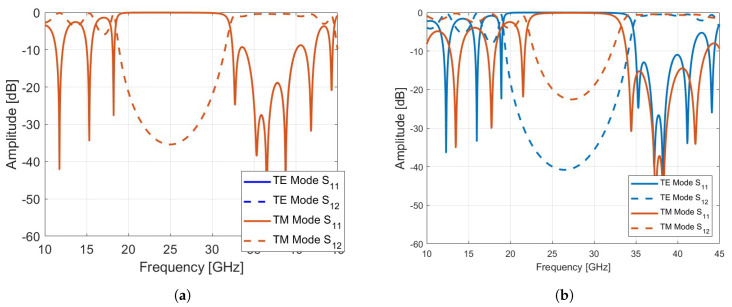
(**a**) The reflection and transmission coefficient under normal incidence. (**b**) The reflection and transmission coefficient under oblique incidence $40^{\circ }$.

**Figure 4 sensors-24-06512-f004:**
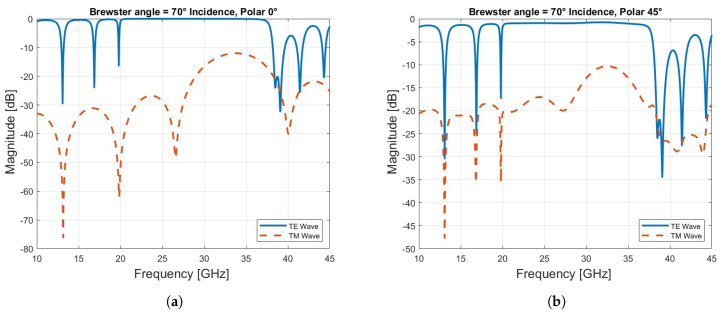
(**a**) The reflection and transmission coefficient at Brewster angle. (**b**) The reflection and transmission coefficient at Brewster angle with 45° linear polarization.

**Figure 5 sensors-24-06512-f005:**
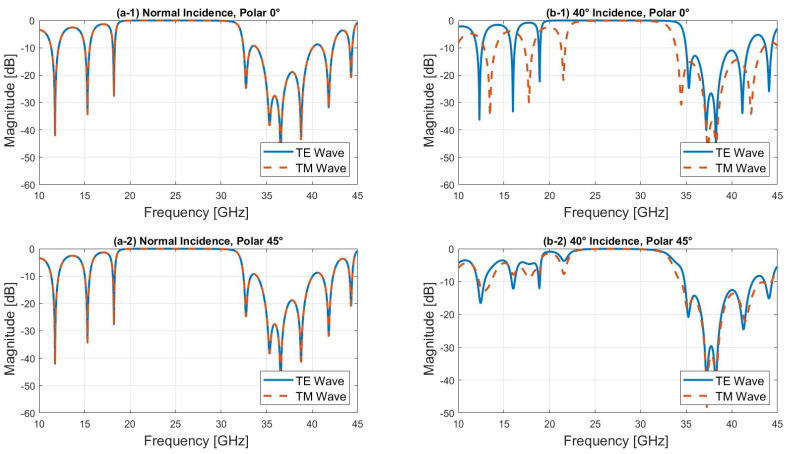
Reflection coefficients of the TE and TM modes at θ = 0° and 40°.

**Figure 6 sensors-24-06512-f006:**
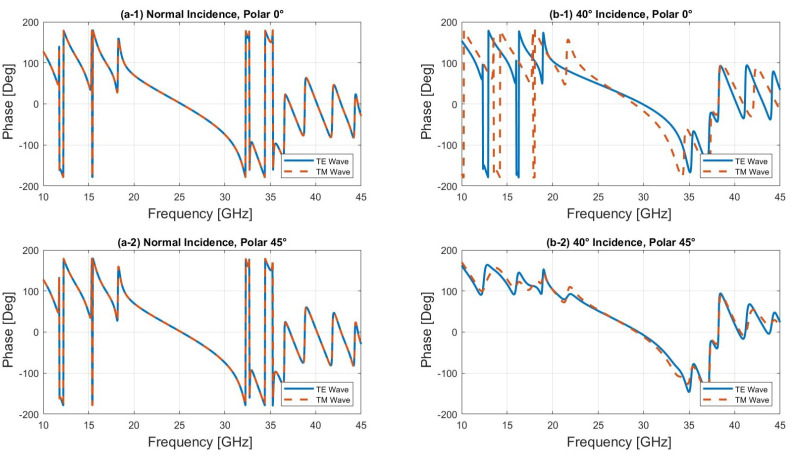
Phase of the TE and TM modes at θ = 0° and 40°.

**Figure 7 sensors-24-06512-f007:**
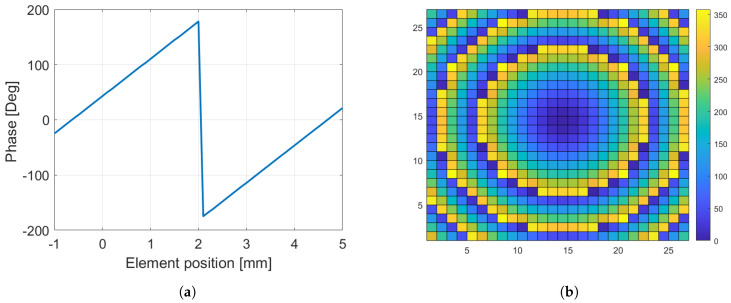
(**a**) Phase curve versus element position along the z−axis. (**b**) Phase distribution on the reflectarray aperture.

**Figure 8 sensors-24-06512-f008:**
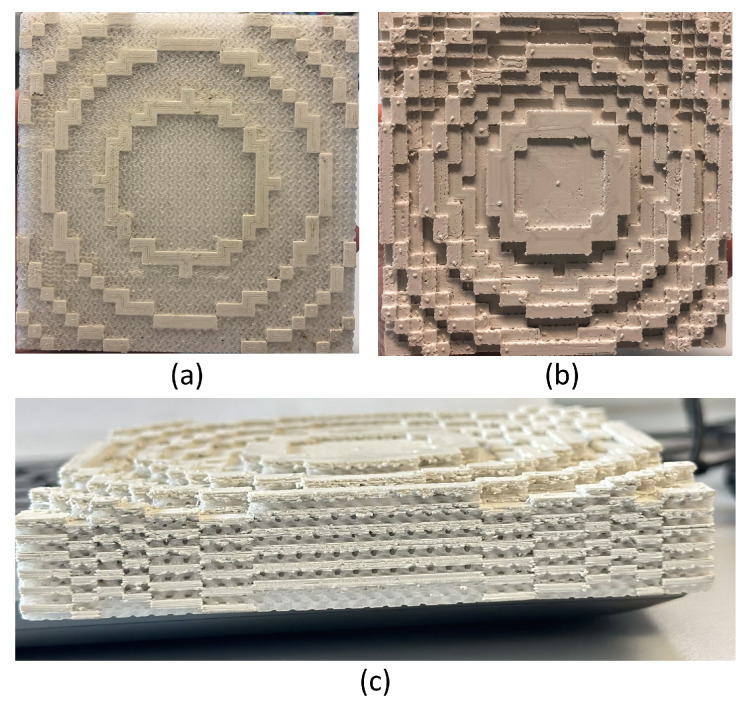
(**a**) Front view of the structure. (**b**) Back view of the structure. (**c**) Side view of the structure.

**Figure 9 sensors-24-06512-f009:**
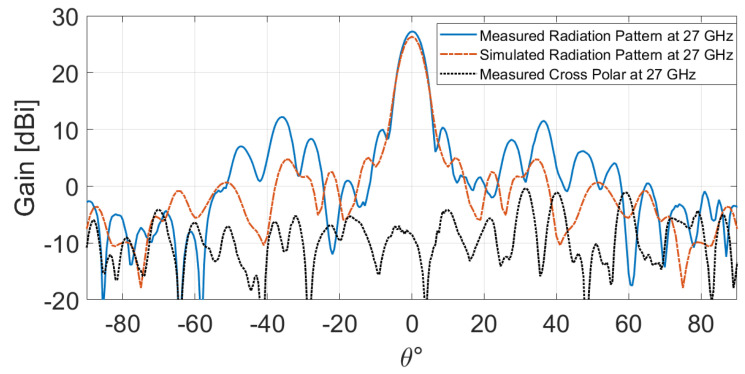
Simulated and measured radiation patterns at 27 GHz.

**Figure 10 sensors-24-06512-f010:**
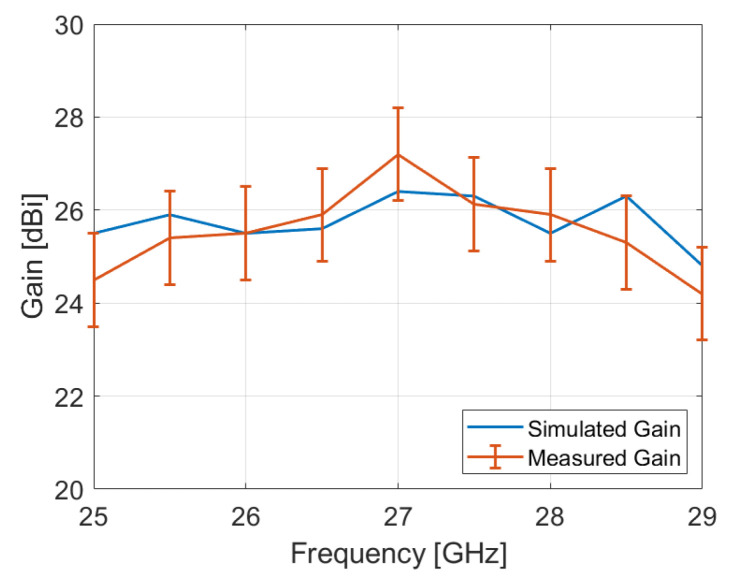
Simulated and Measured Gain over frequency.

**Figure 11 sensors-24-06512-f011:**
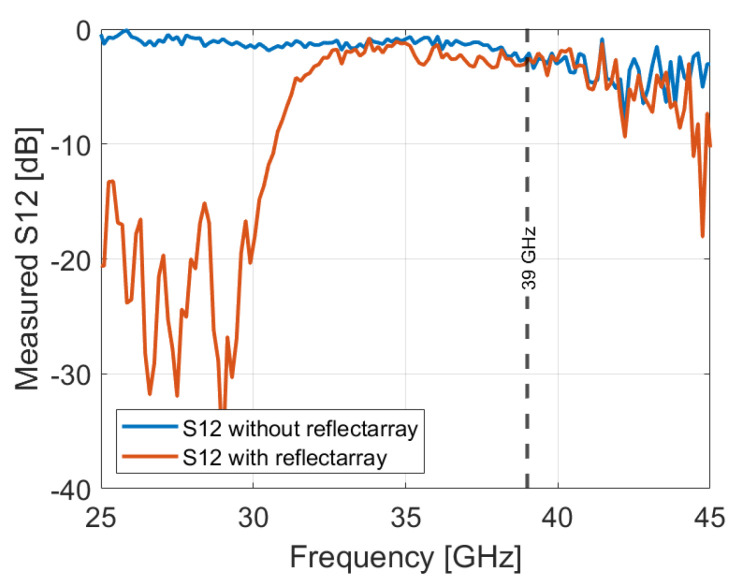
Transmission measurement at 39 GHz.

**Table 1 sensors-24-06512-t001:** Loss budget for the proposed reflectarray at 27 GHz.

Factor	Loss (dB)
Taper efficiency	0.1
Spillover loss	2.7
Blockage loss	0.56
Dielectric loss	0.1

**Table 2 sensors-24-06512-t002:** Comparison of different dual-band reflectarrays.

Ref	[3]	[9]	[12]	[27]	[20]	This Work
Config	DR + Full ground	Multilayer phasing element + FSS	Bragg mirror	Shared aperture phasing element	Dielectric slabs + Full ground	Bragg mirror
Frequency (GHz)	10/5.68	28.8/20	65/24	29.7/19.7	30	39 (future work)/27
Transparency	not transparent	+/− 0.9 dB	not measured	not transparent	not transparent	+/− 0.32 dB
f/d ratio	0.8	0.75	1.94	0.82	1.08	0.5
Size (mm)	S.A 350 × 350	S.A 400 × 400	S.A 77 × 77	C.A r = 194.4	C.A r = 60	S.A 121.5 × 121.5
Gain (dBi)	23/19	38.5/36.3 (Directivity)	30.7/23.2	38.5/35.1	23.66	27.22
Aperture efficiency (%)	11.64/14.37 (estimated)	57/48	33.54/43.75 (estimated)	48.34/50.22 (estimated)	16.32 (estimated)	35.05
Fabrication	N.A	PCB	3D printing	PCB	3D printing	3D printing

S.A: Square Aperture, C.A: Circular Aperture.

## Data Availability

The original contributions presented in the study are included in the article. Further enquiries can be directed to the corresponding author.

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
