# Peer review of "A 3D-Printed Bi-Material Bragg-Based Reflectarray Antenna"

_sensors, 2024, doi:10.3390/s24206512_

Round 1

Reviewer 1 Report

Comments and Suggestions for Authors

This article innovatively presents a reflectarray using bi-material 3D printing with a compact profile for potential applications in 5G communications. Here are some questions for the author to revise and answer:

1.      The authors used 2-bit quantized phase for array cell placement, but did not show a phase distribution map for the array.

2.      Given a z reference plane, the authors control the phase distribution of the cell by varying the distance of the cell from the z plane, the authors also change the thickness of the material to control the phase distribution of the cell? In Fig. 5(c), the thickness of the bottommost dielectric block does not look exactly the same.

3.      In Fig. 7 the authors give a plot of test gain versus simulated gain, however the frequency range is only 25-29 GHz. In the unit design section, the authors discuss transmission and reflection coefficients in the 10-45 GHz range and point out that there is a band gap in the 18-31 GHz range, so why not choose to measure the antenna's gain at 18-31 GHz?

4.      Please show the result of 1dB gain bandwidth.

5.      The authors give a plot of transmission coefficient measurements from 25-45 GHz in Fig. 8, which verifies that the reflectarray has good reflection performance at 25-30 GHz. And the transmitarray performance at 33-42 GHz array is good, do the authors want to show that the designed reflectarray also functions as a transmitarray in this frequency band?

6.      The authors completed the physical fabrication using a 3D printer with dual extruders. Please show more details of the printing process.

Reviewer 2 Report

Comments and Suggestions for Authors

Line 50: Use capslock to initiate phases.

Section 02: Provide the S21 curves for oblique incidence. This allows us to easily see the cell's performance. Did the analysis consider the TE and TM modes? What is the cell's performance under polarization changes?

Reviewer 3 Report

Comments and Suggestions for Authors

The manuscript presents a 3D printed dielectric reflectarray with band gap characteristics for multi-band applications. The band gap characteristics are achieved through the use of photonic crystals made from two dielectric materials with different permittivity. By appropriately selecting the parameters of layered structure, the band gap can be shifted to the desired frequency band, and a good trade-off between transparency and reflectarray thickness can be achieved. The paper presents the characteristics of a 121.5 mm × 121.5 mm reflectarray, fabricated using 3D printing. Table 2 shows a comparison with other dual-band reflectors.  The article can be accepted for publication after minor shortcomings are corrected:

1. It is necessary to improve the quality of Fig. 3 in such a way as to eliminate the stepped lines and transitions between colors on the palettes.

2. In equation (3), f represents not the porosity level but the volume fraction of epsilon2 phase in the mixture.

3. It is necessary to add a comment to Figure 3, in which it is necessary to indicate the polarization state of the incident radiation (s- or p- polarization).

Round 2

Reviewer 2 Report

Comments and Suggestions for Authors

I would like to congratulate the authors for the work.